# Floral transcriptomes reveal gene networks in pineapple floral growth and fruit development

Lulu Wang[1], Yi Li[1], Xingyue Jin[1], Liping Liu[1], Xiaozhuan Dai[1], Yanhui Liu[1], Lihua Zhao[1], Ping Zheng[1], Xiaomei Wang[2], Yeqiang Liu[2], Deshu Lin[1] & Yuan Qin [1,3 ✉]

Proper flower development is essential for sexual reproductive success and the setting of fruits and seeds. The availability of a high quality genome sequence for pineapple makes it an excellent model for studying fruit and floral organ development. In this study, we sequenced 27 different pineapple floral samples and integrated nine published RNA-seq datasets to generate tissue- and stage-specific transcriptomic profiles. Pairwise comparisons and weighted gene co-expression network analysis successfully identified ovule-, stamen-, petal- and fruit-specific modules as well as hub genes involved in ovule, fruit and petal development. In situ hybridization confirmed the enriched expression of six genes in developing ovules and stamens. Mutant characterization and complementation analysis revealed the important role of the subtilase gene *AcSBT1.8* in petal development. This work provides an important genomic resource for functional analysis of pineapple floral organ growth and fruit development and sheds light on molecular networks underlying pineapple reproductive organ growth.

[1] Key Laboratory of Genetics, Breeding and Multiple Utilization of Crops, Ministry of Education, Fujian Provincial Key Laboratory of Haixia Applied Plant Systems Biology, College of Life Sciences, College of Horticulture, Fujian Agriculture and Forestry University, Fuzhou 350002, China. [2] Horticulture Research Institute, Guangxi Academy of Agricultural Sciences, Nanning Investigation Station of South Subtropical Fruit Trees, Ministry of Agriculture, Nanning 530007, China. [3] State Key Laboratory for Conservation and Utilization of Subtropical Agro-Bioresources, Guangxi Key Lab of Sugarcane Biology, College of Agriculture, Guangxi University, Nanning 530004, China. ✉email: yuanqin@fafu.edu.cn

Pineapple (*Ananas comosus* (L.) Merr.) is a perennial monocot belonging to the Bromeliaceae family[1]. Pineapple is the most economically valuable crop utilizing crassulacean acid metabolism and the second most important tropical fruit after banana in terms of international trade[2]. Pineapple is grown in the tropics and warm subtropics worldwide and is the only species of the Bromeliaceae family grown commercially for its fruit. The fleshy multiple fruit of pineapple, referred to as a "sorosis" (syncarp), is derived from the fusion of the ovaries, floral parts, and receptacles of many flowers and is topped by a leafy stem, i.e., the crown[3]. Pineapple is prepared fresh, cooked, juiced, and canned, and the juice can be used to make wines and beers[4]. In addition to its exceptionally palatable and juicy fruit, it has outstanding nutritional and medicinal properties.

Pineapple has a terminal inflorescence that typically consists of 50–200 individual flowers. Each flower contains three sepals, three petals, six stamens, and a gynoecium with a three-carpel inferior ovary[5]. The stigma is trifid, and the style forms a narrow and compact tube structure that is longer than the stamens[6]. Previous studies have investigated the molecular basis of the floral transition in pineapple[7,8], and several genes such as *FLOWERING LOCUS T* (*FT*), *LEAFY* (*LFY*), *PISTILLATA* (*PI*), *FT-LIKE*, and *AP1-LIKE* (*APETALA1*) have been shown to regulate floral meristem identity and floral morphogenesis[9–11]. However, the molecular mechanisms underlying pineapple floral organ development have yet to be characterized. The fully sequenced pineapple genome[12] and other high quality genomicresources[13] available for this species make it an ideal system to study the complex molecular regulatory networks of floral organ growth and fruit development.

Revealing the spatio-temporal gene expressional profile along floral organ growth and fruit development aids in the understanding of the mechanism of reproductive development. Over the years, numerous studies have been implemented to detect the transcriptome profiling of developing petals, ovules, stamens, and fruits/seeds in *Arabidopsis thaliana*[14,15], rice (*Oryza sativa*)[16], soybean (*Glycine max*)[17], tomato (*Solanum lycopersicum*)[18], and maize (*Zea mays*)[19,20]. Although informative, these studies did not provide a complete set of spatio-temporal resolution transcriptome data for the continuously developing reproductive organs in the studied plant species. In this study, we performed transcriptome analysis for 27 different pineapple floral samples, which include three development stages of petal samples, 4 development stages of sepal samples, 7 development stages of gynoecium samples, 7 development stages of ovule samples, and 6 development stages of stamen samples. In combination with previously published datasets for nine pineapple samples, including seven development stages of fruit samples, we performed weighted gene co-expression network analysis (WGCNA) and K-means clustering to identify network modules and tissue-specific gene clusters. Multiple heat shock proteins were identified as hub genes in the Ovule 4-specific module. Genes associated with metabolic processes, including aromatic amino acid family metabolic/catabolic processes, indole derivative metabolic/catabolic processes and tryptophan metabolic/catabolic processes, were found to be enriched in the fruit-specific modules. In addition, subtilisin-like serine protease genes were highly correlated with petals. Our subsequent analysis of the *AcSBT1.8* subtilase gene (Aco003976) showed that it plays a vital role in petal development. The transcriptome and co-expression network analysis described in this study reports a comprehensive high spatio-temporal-resolution of genome-scale gene expression profiling. The findings of this study provide a foundation for the functional analysis of genes involved in pineapple flower and fruit development.

## Results

**Global RNA-seq analysis of pineapple tissues.** To obtain gene-expression profiles for different stages of pineapple flower development, RNA sequencing (RNA-seq) data were generated from 27 different pineapple floral samples [sepal stage 1–4, petal stage 1–3, stamen stage 1–6, gynoecium stage 1–7 (without ovules), and ovule stage 1–7] and each sample has 3 replicates (Fig. 1, Table 1). We also downloaded nine previously published pineapple RNA-seq datasets for Root (mature plants), Leaf (the middle section of the youngest physiologically mature leaf, fourth from the apex), Flower (mixed flowers from different stages), Fruit stage 1–5 and Fruit stage 7[12]. The format "tissue_stage" was used for all sample names; for example, Sepal 1 (abbreviated as Se1) refers to the sample corresponding to the first sepal development stage. For the data generated in this study, the average number of raw reads per library was approximately 45 million, and the average alignment was approximately 65% (Supplementary Fig. 1A, Supplementary Data 1). For the downloaded published data (one replicate provided for each sample), the average number of raw reads per sample and the average alignment were approximately 25 million and 80.7%, respectively[12]. The lower number of raw reads and higher alignment percentage compared to the data generated in this study reflect the use of single-end sequencing for the previously published data. In total, the percentage of uniquely mapped reads was approximately 95%, and the multiple mapping percentage was approximately 5% (Supplementary Fig. 1B), indicating the high quality of the RNA-seq data generated in this study.

Normalized read counts (fragments per kilobase of exon per million reads mapped, FPKM) for each gene were calculated, and genes with FPKM values lower than 0.5 were considered not expressed[21,22]. The pineapple genome is predicted to have 27,024 genes[12]. In our analysis, Fruit 7 had the lowest number of expressed genes (16,157), and the Root sample had the highest number (17,807) (Supplementary Fig. 2A, Supplementary Data 2). All the expressed genes were divided into five categories according to their FPKM values[23]. The moderately expressed genes $(10 \leq FPKM < 100)$ were the majority in all samples (Supplementary Fig. 2A).

The cluster dendrogram analysis (Supplementary Fig. 2B) showed good correlation among the three biological replicates of each sample, except that the Ov2-1 grouped with the Ov1 samples and the Se3-1 grouped with the Se4 samples, which may be due to the high similarity of the samples at the developmental stages very close to each other. These results indicate the reliability of the RNA-seq data generated in this study.

**Construction of gene co-expression networks using the pineapple RNA-seq transcriptome data.** WGCNA is a systems biology approach aimed at understanding networks instead of individual genes. In this study, co-expression networks were constructed based on pairwise correlations of gene expression trends across all sampled tissues. A total of 19,832 differentially expressed genes between any of the two samples among the 36 different samples of pineapple were identified by Cuffdiff and edgeR. The gene-expression profiles of all these 19,832 genes were analyzed to identify gene co-expression modules using R package WGCNA[24]. The matrix was raised to a soft-thresholding power ($\beta = 11$ in this study) (Fig. 2a). Modules are defined as clusters of highly interconnected genes, such that genes within the same cluster have high correlation coefficients with one another. For the present analysis, the minimum module size was set to 30, and modules with highly correlated eigengenes (based on a threshold of 0.25) were merged; an eigengene is defined as the first principal

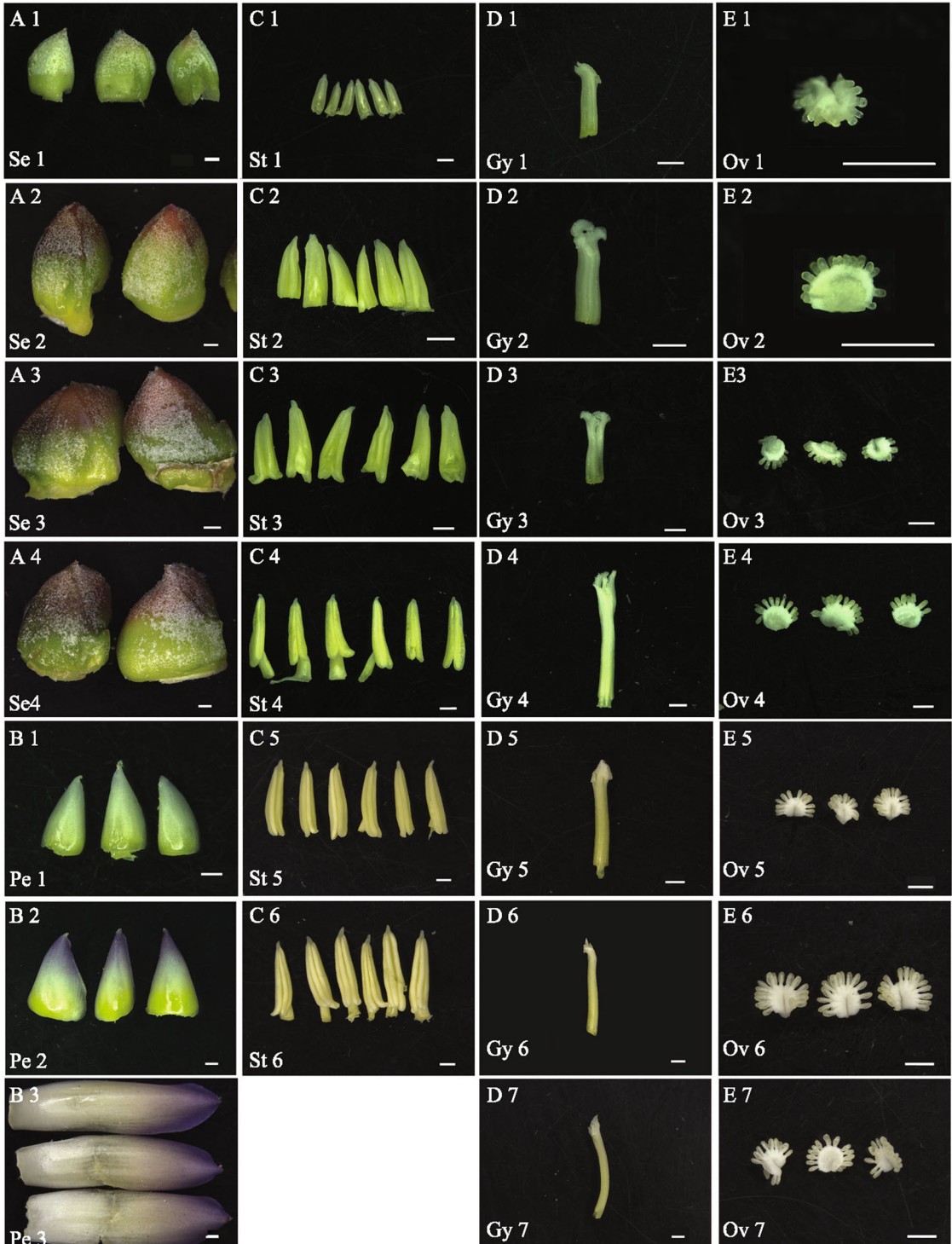

**Fig. 1 Morphological characteristics of the pineapple floral tissues used for RNA-seq analysis. a** Sepal samples at 4 different stages. **b** Petal samples at three different stages. **c** Stamen samples at six different stages. **d** Gynoecium samples without ovules at seven different stages. **e** Ovule samples at seven different stages. Bar = 1 mm.

component of a given module and may be considered as a representative of the gene-expression profiles in that module. Twenty distinct modules were identified (Fig. 2b), excluding the gray module, which was used to hold all genes that did not clearly belong to any other module. The modules are labeled by color and are shown in a hierarchical clustering dendrogram, in which each tree branch constitutes a module and each leaf in the branch is one gene (Fig. 2c, Supplementary Data 3).

Next, we performed a correlation analysis between the 20 distinct modules and the 36 tissues/stages (Fig. 2d). The colored shading in the chart indicates significant correlations between the indicated modules and samples (red and blue corresponding to positive and negative correlation, respectively) based on the tissue-specific expression profiles of the eigengenes (Fig. 2d, Supplementary Data 4). Eight of the 20 modules had a significant positive correlation with a single tissue/stage ($r > 0.8$, $P < 10^{-3}$,

**Table 1 Description of the 27 pineapple floral samples used for RNA-seq analysis.**

| Sepal stages | Petal stages | Stamen stages | Gynoecium stages | Ovule stages | Flower characteristics |
|---|---|---|---|---|---|
| – | – | St1 | Gy1 | Ov1 | Bud width < 5 mm |
| – | – | St2 | Gy2 | Ov2 | 5 mm ≤ Bud width < 8 mm |
| Se1 | Pe1 | St3 | Gy3 | Ov3 | Bud width = 8 mm |
| – | – | St4 | Gy4 | Ov4 | Bud width > 8 mm to petal invisible |
| Se2 | Pe2 | St5 | Gy5 | Ov5 | Petal just visible |
| – | – | St6 | Gy6 | Ov6 | 1 mm petal visible |
| – | – | – | Gy7 | Ov7 | >2 mm petal visible to pre-flowering |
| Se3 | Pe3 | – | – | – | Flowering |
| Se4 | – | – | – | – | After flowering |

*Se* sepal, *Pe* petal, *St* stamen, *Gy* gynoecium, *Ov* ovule.

Fig. 2d, Supplementary Data 4). For example, the lightcyan 1, yellow, green, plum 1, and sienna 3 modules were specifically correlated with Ovule 4, Petal 3, Root, Petal 1, and Gynoecium 4, respectively. Notably, some modules were highly related to a specific tissue throughout the developmental process of that tissue. For instance, brown was associated with all ovule stages and light yellow was positively correlated with all fruit stages (Fig. 2d).

**Three modules were positively correlated with ovule development**. The brown, yellowgreen, and lightcyan 1 modules were positively correlated with ovule development (Fig. 2d). The brown module was the largest module with 4070 eigengenes. GO-enrichment analysis for eigengenes in this module identified many meiosis or mitosis-related GO categories, such as chromosome organization, chromosome condensation, double-strand break repair, mitotic sister chromatid segregation, mitotic cell cycle, DNA replication, and so on (Supplementary Data 5). In line with the active cell division taken place during ovule development, GO terms associated with kinesin family proteins, tubulin-binding proteins, and microtubule-associated proteins were also significantly enriched in the brown module (Supplementary Data 5).

To identify the potential important genes involved in ovule development, module eigengene connectivity (kME) was calculated for each gene within the brown module (Supplementary Data 3). A kME value of 1 indicates perfect correlation with the module eigengene. Genes with a high kME (kME > 0.9) are considered hub genes or connection hubs in the network, which are expected to play important roles in the biological processes. A total of 360 hub genes were identified in the brown module (Supplementary Data 6). The top-ranked hub gene was Aco018816, which encodes a mediator of DNA damage checkpoint protein. GO analysis among the 360 hub genes showed an enrichment of genes associated with DNA metabolic process, nucleic acid metabolic process, heterocycle metabolic process, organic cyclic compound metabolic process, and DNA replication (Supplementary Fig. 3A). Twenty-nine of them were transcriptional factor (TF) genes (Supplementary Data 6), including one FAR1 family TFs (Aco028201), four B3 TFs (Aco004015, Aco012568, Aco009528, and Aco025803), and two GRF TFs (Aco000277 and Aco013172). The orthologous genes of these TFs (FAR1 and AT5G18960[25]; B3, AT5G42700, AT5G58280, AT3G19184, AT2G24700[26]; GRF, AT2G22840, and AT4G37740[27]) in *Arabidopsis* have been reported to play essential roles in ovule and flower development. The functions of these genes in pineapple ovule development were worth being investigated in future studies.

The correlation of the yellowgreen eigengene with ovules increased with the successive stages of ovule development (Fig. 2d;

Supplementary Data 4, Supplementary Fig. 3B). Transcript abundance for this module also increased as ovule development progress. GO analysis for eigengene in the yellowgreen module showed an enrichment of genes associated with peptidase activity, endopeptidase activity, serine hydrolase activity, ovule development, and cell wall assembly (Supplementary Data 5). Of the 28 hub genes (kME > 0.90) identified in the yellowgreen module, the top-ranked hub gene was Aco021283 (kME > 0.99), which encodes a pathogenesis-related thaumatin superfamily protein and expressed specifically in developing ovule (Supplementary Data 6), suggesting its potential role in ovule development. Notably, four (Aco000421, Aco001204, Aco024518, and Aco015216) out of the 28 hub genes in this module belong to the proteins of unknown function DUF239 subfamily, which encodes putative carboxyl-terminal peptidase (Supplementary Data 6). DUF239 family genes in *Arabidopsis*, such as AT4G17505, AT5G25950, and AT4G23350, were highly enriched in the female gametophyte and were proposed to play key roles in ovule development in *Arabidopsis*[28]. It will be of interest to investigate the function of these DUF239 genes in pineapple ovule development. Another hub gene identified in this module was the MADS-box TF gene Aco008359 (Supplementary Fig. 3C). Aco008359 is homologous to *Arabidopsis AGL32*, known to play important role in ovule integument development implicating that Aco008359 could also function in pineapple ovule development[29,30].

The lightcyan1 module was highly correlated with Ovule 4 (Fig. 2d). The eigengenes in the lightcyan1 module were most highly expressed in the Ovule 4 sample (Fig. 3a). GO enrichment analysis indicated that genes associated with nucleotide or ribonucleotide binding were enriched in the lightcyan1 module (Fig. 3d). Many of the eigengenes in this module identified by co-expression network analysis encode heat shock proteins suggesting that the reproductive process at this stage is highly sensitive to stress (Fig. 3g).

**The lightyellow module was highly correlated with fruit development**. Pineapple fruit has outstanding nutritional and medicinal properties, and it is the most important part of this crop species. The lightyellow and salmon modules were both highly correlated with fruit development (Fig. 2d). GO enrichment analysis was performed for eigengenes in these two modules, but no enriched terms were identified for the salmon module, which had a small number of hub genes (15 genes). Most eigengenes in the lightyellow module were annotated with metabolic process terms, such as aromatic amino acid family metabolic/catabolic process, indole derivative metabolic/catabolic process, and tryptophan metabolic/catabolic process (Supplementary Data 5; Supplementary Fig. 4). These processes may be important for different aspects of the taste of pineapple fruit, such

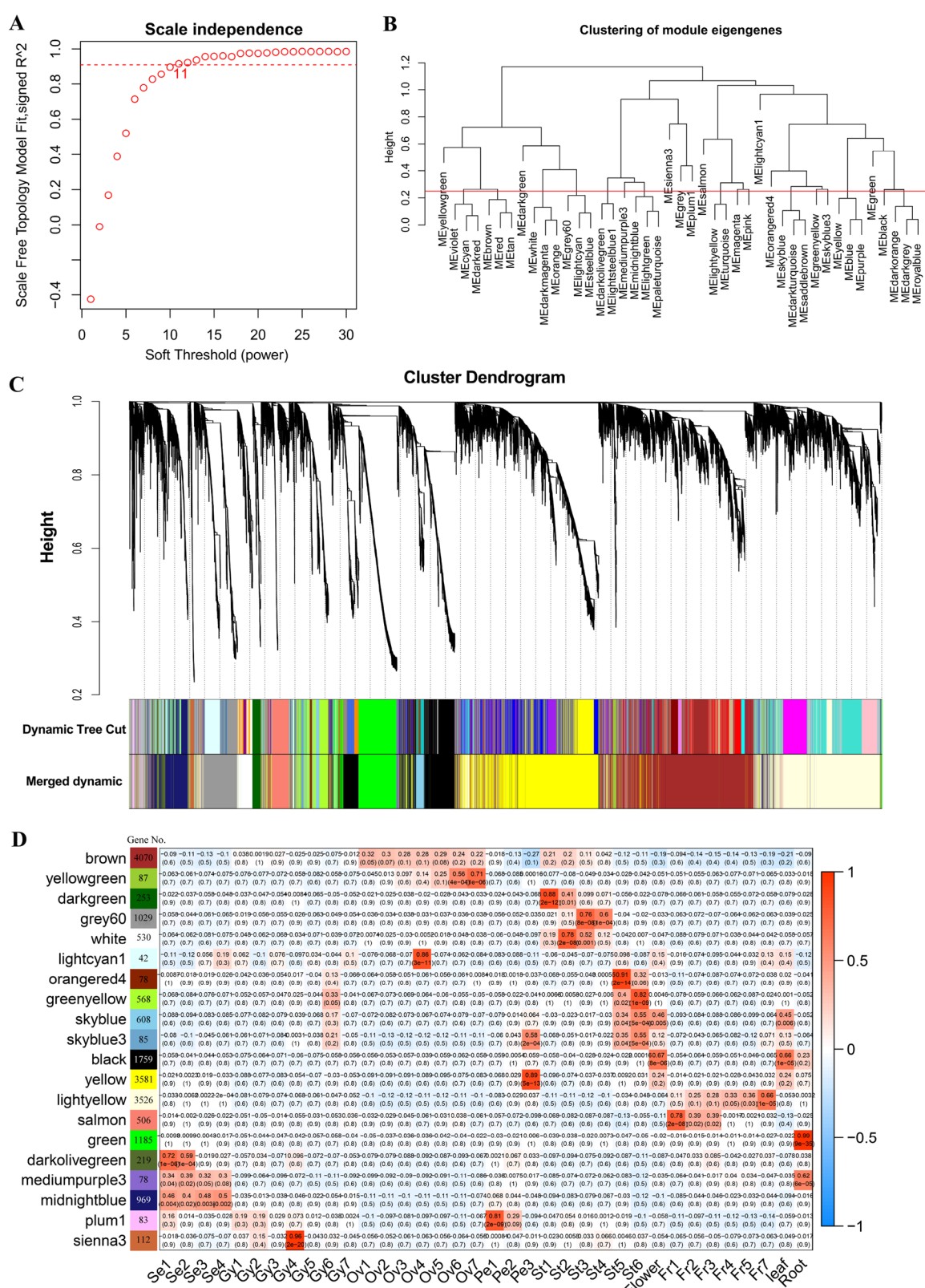

as sweetness and flavor. Three kynurenine formamidase family genes (Aco018143, Aco030096, and Aco026349) were the top genes associated with these GO terms (Supplementary Data 5). Ten TF genes were among the lightyellow hub genes, and it will be interesting to determine whether and how these genes regulate pineapple fruit development and flavor formation: two WRKY

genes (Aco000069 and Aco002567), one NAC gene (Aco018555), three HB:TALEs (Aco012803, Aco015871 and Aco008183), one bZIP gene (Aco001167), one bHLH gene (Aco005342) and two AP2/ERFs (Aco009511 and Aco021063) (Supplementary Table 1). The orthologs of the pineapple bZIP and bHLH family TFs in tomato and strawberry, such as *SlbZIP1* and *SlbZIP2*[31],

**Fig. 2 Weighted gene co-expression network analysis for pineapple RNA-seq data. a** Analysis of network topology for different soft-thresholding powers. The graph displays the influence of soft-thresholding power ($x$-axis) on the scale-free fit index ($y$-axis). **b** Cluster dendrogram of module eigengenes. Branches of the dendrogram group together eigengenes that are positively correlated. The red line is the merging threshold, and groups of eigengenes below the threshold represent modules whose expression profiles should be merged due to their similarity. **c** Hierarchical cluster dendrogram showing co-expressed modules identified by weighted gene co-expression network analysis for the pineapple RNA-seq data. Each leaf on the tree represents one gene. The major tree branches constitute 20 merged modules (based on a threshold of 0.25), labeled with different colors. **d** Module-tissue association analysis. Each row corresponds to a module, with the number of genes in the module indicated on the left. Each column corresponds to a specific tissue. The correlation coefficient between a given module and tissue type is indicated by the color of the cell at the row-column intersection. Red and blue indicate positive and negative correlation, respectively.

*SlbHLH22*[32], and *FaSPT*[33] have been reported to play key roles in fruit development. The functions of these genes in pineapple fruit development required further investigation.

**The plum1 and yellow modules were highly correlated with petal development**. The module-tissue association analysis indicated that the plum1 and yellow modules were highly correlated with Petal 1 and Petal 3, respectively (Fig. 2d). Moreover, the eigengenes in these modules were most highly expressed at the Petal 1 and Petal 3 samples, as shown by the heatmap diagrams (Fig. 3b, c). GO enrichment analysis indicated that the GO terms with peptidase and proteolysis activity were enriched among the plum1 module (Petal 1) eigengenes, and six subtilisin-like serine protease family genes (Aco022840, Aco018854, Aco025812, Aco000451, Aco005467, and Aco005469) in this module were annotated with these terms (Fig. 3e, Supplementary Data 3). Notably, four (Aco005469, Aco018854, Aco000451, Aco025812) out of these six genes were also identified as connection hubs in the plum 1 module network (Fig. 3h, Supplementary Data 6), indicating the potential involvement of subtilase family proteins in early pineapple petal development[34]. The enriched GO terms among the yellow module (Petal 3) genes included hydrolase activity and transmembrane transporter activity (Fig. 3f). Interestingly, this module also included 11 subtilase family genes among its hub genes (Fig. 3i, Supplementary Data 6), all of which were distinct from the subtilase genes associated with the Petal 1 module. In addition, the 2 groups of subtilase genes had opposite expression patterns over the course of petal development: the expression of the 6 Petal 1-associated subtilase genes gradually decreased, while the expression of the 11 Petal 3-associated subtilase genes gradually increased (Supplementary Fig. 5), indicating that these subtilase genes may have been diversified.

**Identification of spatiotemporal expression trends across pineapple transcriptomes by K-means clustering**. In addition to WGCNA, we used K-means clustering to identify co-expressed gene clusters. There are totally 21 modules identified by the gene co-expression analysis, thus the number of K-means clustering was set to 21, resulting in identification of 21 K-means clusters (Supplementary Fig. 6A). Seventeen of the 21 K-means clusters included 15,872 genes with distinct stage- and tissue-specific expression patterns (Fig. 4a). Clusters with similar expression trends in the 17 clusters were further combined into seven superclusters, each with a unique gene expression profile across the analyzed tissues and stages (Fig. 4a). Ovules-specific clusters (C2, C3, and C5) and fruit-specific gene clusters (C9, C16, and C17) corresponded to the largest two modules identified by WGCNA, brown and lightyellow module, which had a highly positive association with ovule and fruit tissues, respectively (Fig. 2d). A comparison between tissue-specific genes from WGCNA and K-means revealed the consistency between the results obtained from two methods. For example, of the 4157 eigengenes in ovule modules (brown and lightyellow) identified

by WGCNA, 3216 genes (77.4%) were also detected in the ovules-specific clusters (C2, C3, and C5) by K-means (Supplementary Fig. 6B). Of the 1559 eigengenes in stamen modules (grey60 and white) identified by WGCNA, 978 genes (62.6%) were also detected in the steman 1–4 specific cluster (C1 and C11) (Supplementary Fig. 6B). Interestingly, stamens at early and late development stages were associated with different clusters. Moreover, three small modules identified by WGCNA, sienna3 (113 genes), lightcyan1 (43 genes), and plum1 (84 genes), which were highly correlated with Gynoecium 4, Ovule 4 and Petal 1, respectively (Fig. 2d), were not detected by K-means clustering. This difference suggests that WGCNA is a more powerful tool for identifying groups of highly correlated genes that co-occur across different pineapple tissues/stages and for constructing co-expression gene networks.

Of the 15,872 genes in the seven superclusters, 918 were transcription factors (TFs) genes representing 66 families and 60 were transcription coregulators (TCs) (Supplementary Data 7)[35]. These TF genes had distinct stage- and tissue-specific expression patterns (Fig. 4b, Supplementary Data 7), and their dynamic expression changes may be informative about their functions. For example, genes belonging to the AP2, B3, bHLH, bZIP, C2C2, MYB, and NAC TF families had high ovule-specific expression and belonged to supercluster 1 (Supplementary Data 7), suggesting that these TFs may play important roles in ovule development. In line with this results, *Arabidopsis AIL5* (AT5G57390), and *ARF3* (AT2G33860) and *KAN2* (AT1G32240) genes, the orthologs of the pineapple gene Aco014708, and Aco021382 and Aco013194 belonging to the AP2/ERF:ERF, B3:ARF and GARP: G2-like TF family in supercluster 1 (Supplementary Data 8), which were also identified as eigengenes of the ovule (brown) module by WGCNA (Supplementary Data 3), were reported to play important roles in *Arabidopsis* ovule development[36–39]. MYB and GRAS TFs have been shown to involve in fruit development in black wolfberry and strawberry[40,41]. Notably, 19 MYB and 13 GRAS TFs were identified in the fruit-specific supercluster 2, suggesting the potential role of these MYB and GRAS TFs in regulating pineapple fruit development.

**Pineapple ABCE genes**. The floral homeotic genes, which constitute the ABCE model, are highly expressed in flower tissues and are well-studied genes involved in flower development[42]. To understand the molecular control of floral organ patterning in pineapple, we identified homologs of the ABCE genes in pineapple by sequence alignment and phylogenetic analysis. As a result, 11 ABCE genes, including two class A genes (*AcAP1* and *AcAP2*), three class B genes (*AcAP3a*, *AcAP3b* and *AcPI*), one class C gene (*AcAG*), and five class E genes (*AcSEP1*, *AcSEP2*, *AcSEP3*, *AcSEP4*, and *AcAGL6*) were identified (Table 2). The number of ABCE genes in pineapple is more than *Arabidopsis*, but less than rice ABCE genes (Fig. 5a and Supplementary Data S9). In agreement with their putative roles in floral organ patterning, the expression pattern of the pineapple ABCE genes had higher expression level in floral tissues compared to

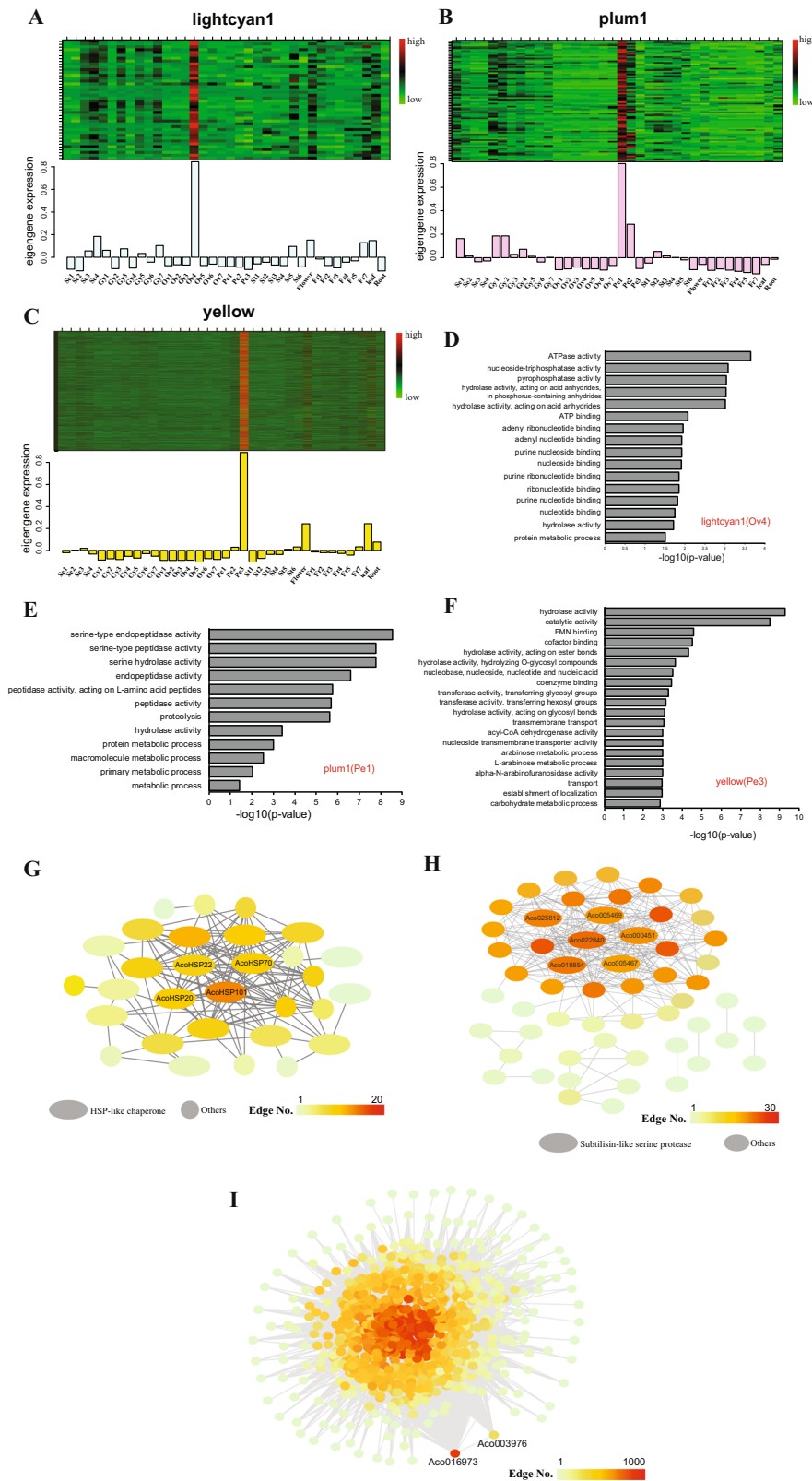

vegetative tissues (Fig. 5b). Class A genes were preferentially expressed in sepal and petal tissues, whereas Class B genes exhibited high expression level in the early developmental stages of petal and late developmental stages of the stamen. These results suggest that *AcAP1* and *AcAP2* may play roles in sepal and sepal development, whereas *AcAP3a, AcAP3b* and *AcPI* may involve in petal and stamen development. Class C gene, *AcAG* was highly

expressed in all stages of gynoecium, ovule and stamen tissues, indicating its potential role in stamen and carpel development. Among the class E genes, *AcSEP1* showed a low expression level with weak expression in late developmental stages of stamen and gynoecium. The expression profiles of *AcAP1*, *AcPI*, and *AcSEP1* and their function in flower organ development were consistent with previous research[9,43,44]. In contrast, *AcSEP2* and *AcSEP3*

**Fig. 3 Genes, enriched GO terms and networks of the Ovule 4-, Petal 1-, and Petal 3-specific modules. a** Eigengene expression profile for the Ovule 4 (lightcyan1) module in different samples. Top panel: expression heatmap showing the relative FPKM of all genes from the Ovule 4 (lightcyan1) module. Bottom panel: the $x$-axis indicates the samples, and the $y$-axis indicates the log2 "relative FPKM values" of the module eigengene. **b** Eigengene expression profile for the Petal 1 (plum1) module in different samples. **c** Eigengene expression profile for the Petal 3 (yellow) module in different samples. **d** Enriched GO terms for the Ovule 4 (lightcyan1) module. **e** Enriched GO terms for the Petal 1 (plum1) module. **f** Enriched GO terms for the Petal 3 (yellow) module. **g** The correlation network of the Ovule 4 (lightcyan1) module. Thirty genes with edge weights greater than 0.3 were included in the Cytoscape-generated diagram; oval shapes were used to indicate the 19 heat shock protein genes. **h** The correlation network of the Petal 1 (plum1) module. Fifty-four genes with edge weights greater than 0.3 were included in the Cytoscape-generated diagram; the ovals indicate the six subtilisin-like serine protease family genes. **i** The correlation network of the Petal 3 (yellow) module. The Cytoscape-generated diagram includes 1031 genes with edge weights greater than 0.4.

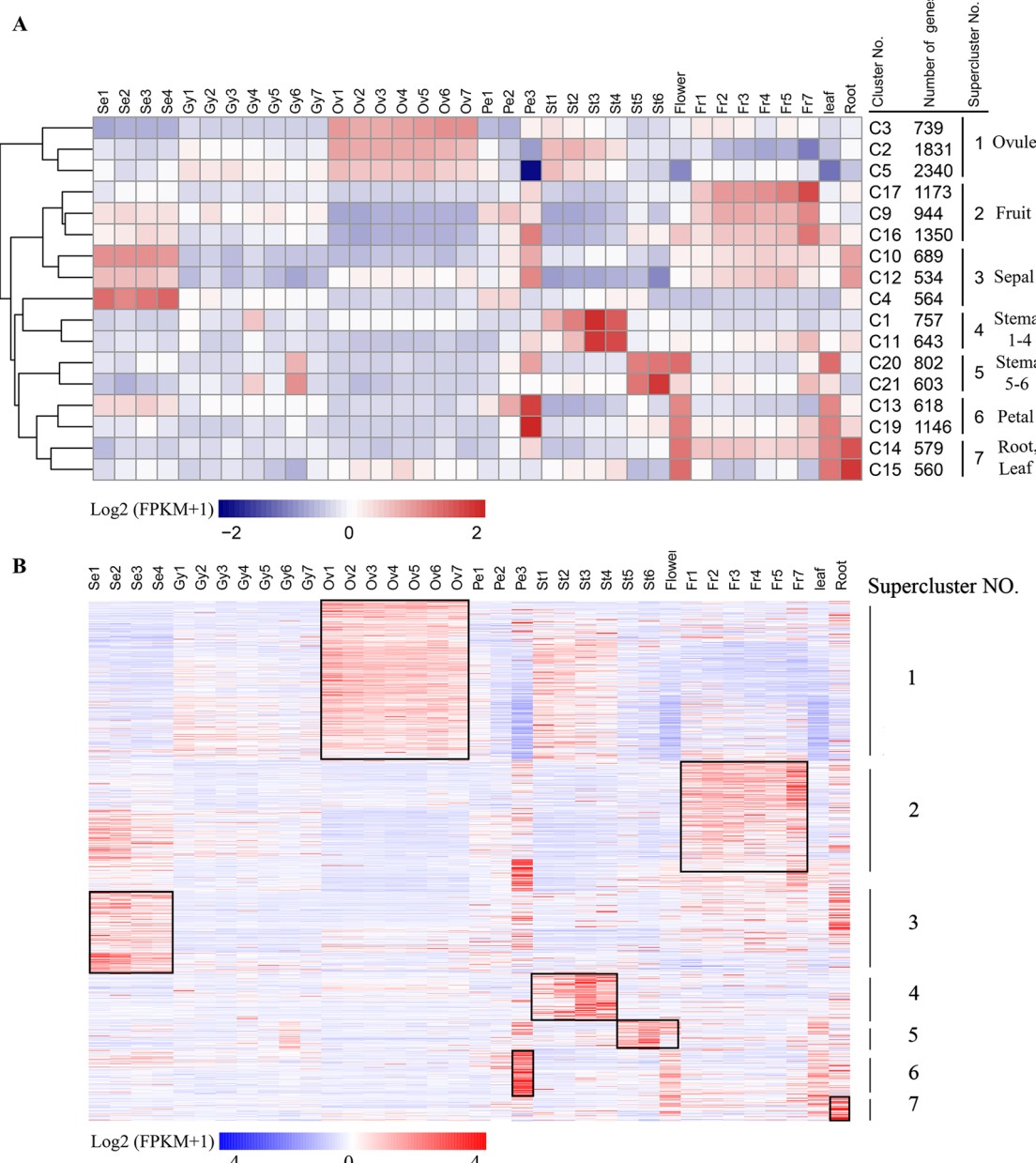

**Fig. 4 Seven superclusters of genes with unique tissue- or stage-specific expression profiles. a** Seventeen of the 21 K-means clusters included 15,872 genes with distinct stage- or tissue-specific expression patterns. The scale represents the Log2(FPKM + 1) values of all genes in a given cluster. Clusters with similar expression trends were combined to form seven superclusters. **b** Heatmap of the Log2(FPKM + 1) values of 918 individual transcription factor (TF) genes. The black boxes show the high expression TFs in each Supercluster.

**Table 2 The relationships of floral homeotic genes among *Arabidopsis*, rice, and pineapple.**

| Class | Arabidopsis | | Rice | | Pineapple | |
|---|---|---|---|---|---|---|
| | Name | Gene ID | Name | Gene ID | Name | Gene ID |
| Class A | AtAP1 | AT1G69120 | OsMADS14 | LOC_Os03g54160 | AcAP1 | Aco012428 |
| | | | OsMADS15 | LOC_Os07g01820 | | |
| | | | OsMADS18 | LOC_Os07g41370 | | |
| | AtAP2 | AT4G36920 | AP2D2 | LOC_Os03g60430 | AcAP2 | Aco012448 |
| Class B | AtAP3 | AT3G54340 | OsMADS16 | LOC_Os06g49840 | AcAP3a | Aco025594 |
| | | | | | AcAP3b | Aco017589 |
| | AtPI | AT5G20240 | OsMADS2 | LOC_Os01g66030 | AcPI | Aco019365 |
| | | | OsMADS4 | LOC_Os05g34940 | | |
| Class C | AtAG | AT4G18960 | OsMADS58 | LOC_Os05g11414 | AcAG | Aco009993 |
| | | | OsMADS3 | LOC_Os01g10504 | | |
| Class E | AtSEP1 | AT5G15800 | OsMADS7 | LOC_Os08g41950 | AcSEP1 | Aco019039 |
| | AtSEP2 | AT3G02310 | OsMADS8 | LOC_Os09g32948 | AcSEP2 | Aco003667 |
| | AtSEP3 | AT1G24260 | OsMADS1 | LOC_Os03g11614 | AcSEP3 | Aco015105 |
| | AtSEP4 | AT2G03710 | OsMADS34 | LOC_Os03g54170 | AcSEP4 | Aco017563 |
| | | | OsMADS5 | LOC_Os06g06750 | | |
| | AtAGL6 | AT2G45650 | OsMADS17 | LOC_Os04g49150 | AcAGL6 | Aco015487 |
| | | | OsMADS6 | LOC_Os02g45770 | | |

highly expressed in all four whorls, especially in petal, stamen, gynoecium, and ovules. However, *AcSEP4* and *AcAGL6* were preferentially expressed in sepal and petals. Altogether, these results suggest that the class E genes in pineapple may have undergone subfunctionalization with *AcSEP2* and *AcSEP3* playing the main role in the inner three whorls, whereas *AcSEP4* and *AcAGL6* are dominant in the outer two whorls.

**Confirmation of gene expression patterns by in situ hybridization and RT-qPCR**. To confirm and further characterize the expression patterns of some of the above tissue- and stage-specific genes, we performed in situ hybridization for three Ovule 4-associated genes encoding the chaperone protein ClpB (Aco005999), a MADS-box TF (Aco008359) and a 70 kDa heat shock protein (Aco001460). All three genes were preferentially expressed in the nucellus of stage 4 pineapple ovules (Fig. 6a). We also performed in situ hybridization for three Stamen 5-enriched genes encoding a class I glutamine amidotransferase-like superfamily protein (Aco005285), an aquaporin-like superfamily protein (Aco008952) and a sequence-specific DNA binding TF (Aco002322). The first of these, Aco005285, was highly enriched in the four non-reproductive layers, namely, the epidermis, endothecium, middle layer, and tapetum, of the four lobes in pineapple stamens at stage 5 (Fig. 6b). The other two genes were mainly detected in the tapetum layer of the stamens at stage 5 (Fig. 6b).

To validate the expression patterns of the subtilase genes in petal tissues, we randomly selected 9 petal-specific expressed subtilase genes (Aco003976, Aco013912, Aco019115, Aco031678, Aco015700, Aco005467, Aco022840, Aco005469, Aco018854), and tested their expression level in different developmental stages of petals by real-time quantitative PCR (RT-qPCR). The RT-qPCR results were consistent with our RNA-seq results (Fig. 6c), suggesting that the petal-specific genes expression patterns based on WGCNA and K-means clustering data are reliable.

**Pineapple *AcSBT1.8* regulates petal development**. We showed above that the subtilase genes were overrepresented in the eigengenes of the Petal 1 and Petal 3 modules. Interestingly, there was no overlap between the Petal 1-associated and the Petal 3-associated subtilase genes, and the two groups of subtilase genes had opposite expression patterns throughout petal development.

The significant enrichment of subtilase genes in petal-specific modules and the distinct expression patterns of the subtilase genes prompted us to investigate its function in petal development. We obtained the *Arabidopsis* mutant lines *sbt1.8* (Salk_020799), *sbt5.3* (Salk_125788), and *sbt5.4* (Salk_025087) corresponding to the *Arabidopsis* homologs of the pineapple subtilase genes Aco003976 (*SBT1.8*) and Aco000861 (*SBT5.3* and *SBT5.4*), respectively. In the *sbt1.8* mutant, the T-DNA insertion in an exon of the gene leads to *AtSBT1.8* knockout (Supplementary Fig. 7A, B). Among the three *Arabidopsis* mutants, only *sbt1.8* had a petal phenotype distinct from wild type (WT). At the mature flower stage (stage 14), the petals of the *sbt1.8* mutant flowers were significantly longer and wider than WT petals (Fig. 7a–d). However, the petal length-to-width ratio of *sbt1.8* was comparable to that of WT (Fig. 7d). In contrast to the increased petal size in the *sbt1.8* mutant, leaf size was comparable in *sbt1.8* and WT (Supplementary Fig. 7g). The larger petal phenotype of the *sbt1.8* mutant was rescued by a construct containing the DNA sequence of the pineapple homolog *AcSBT1.8* (Aco003976) driven by the 35S promoter (Fig. 7a–c). Together, the observed petal defects and phenotypic rescue suggest that *SBT1.8* regulates petal development and that this function is conserved across species.

To examine whether the increased petal size of *sbt1.8* was due to altered cell expansion or cell proliferation, we first analyzed the two-dimensional area at the center of the petal blades in WT and *sbt1.8*. The length and width of the adaxial and abaxial epidermal cells in *sbt1.8* were comparable to those in WT (Supplementary Fig. 7c, d), indicating that *SBT1.8* does not control cell expansion in petals. In contrast, the total cell number in the petal blade region was significantly greater in the *sbt1.8* mutant compared to WT (Fig. 7e), indicating that *SBT1.8* may regulate petal development by inhibiting cell proliferation. It has been reported that petal size is controlled by the SEPALLATA3-regulated miR319-TCP4 module[45]. To determine whether *SBT* regulates petal development through this signaling pathway, we compared the expression levels of *SEP3*, *MIR319a*, and *TCP4* in WT and *sbt1.8*. Comparable expression levels of these genes were detected in WT and *sbt1.8* (Supplementary Fig. 7I), suggesting that *SBT* regulated petal development is likely to be independent of the SEP3-MIR319/TCP4 signaling pathway. Because the petal adaxial epidermis in *Arabidopsis* contains conically shaped cells, we also investigated whether this particular cell shape was affected in the *sbt1.8* mutant. Scanning electron microscopy revealed no obvious

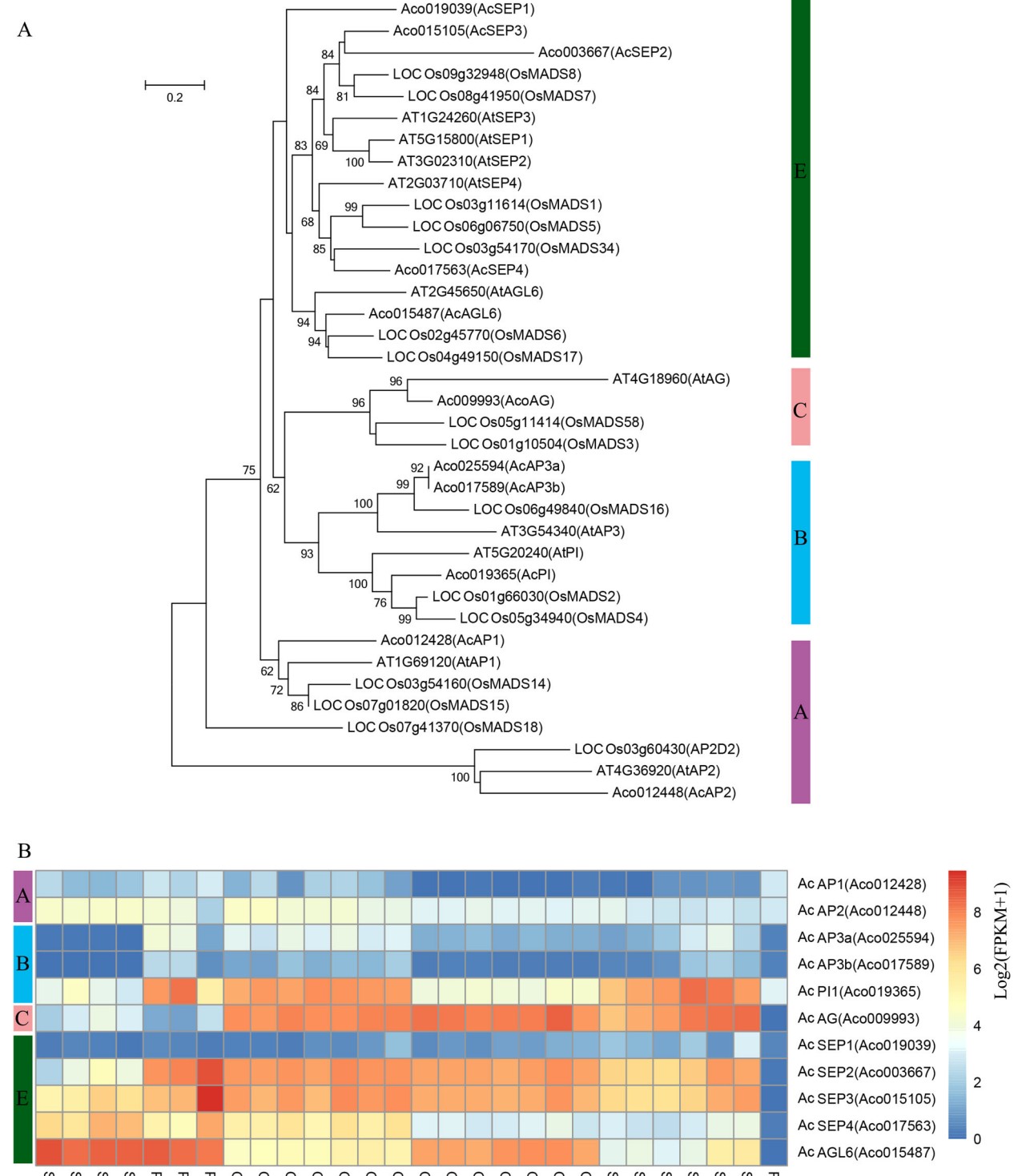

**Fig. 5 Identification of pineapple floral homeotic genes and expression profiles. a** Phylogenetic tree of floral homeotic genes in *Arabidopsis*, rice and pineapple. The full length protein sequences were aligned by Clustal Omega and used for constructing the phylogenetic tree by neighbor-joining in MEGA5. Values above branches were bootstrap percentages (1000 replicates). **b** Expression pattern of A, B, C, and E class genes in pineapple flower and vegetative tissues. Log2(FPKM + 1) values were used for the heatmap.

differences in the conical cell angle between WT and *sbt1.8* (Supplementary Fig. 7E, F), indicating that *SBT1.8* does not regulate this conical cell shape. In addition, we did not observe conically shaped cells in pineapple petals (Supplementary Fig. 7H).

## Discussion

Pineapple, a popular tropical fruit with a unique floral structure and delicate taste, is grown in suitable climates worldwide and is now the most economically important plant in the Bromeliaceae family[46]. The terminal inflorescence of the pineapple plant

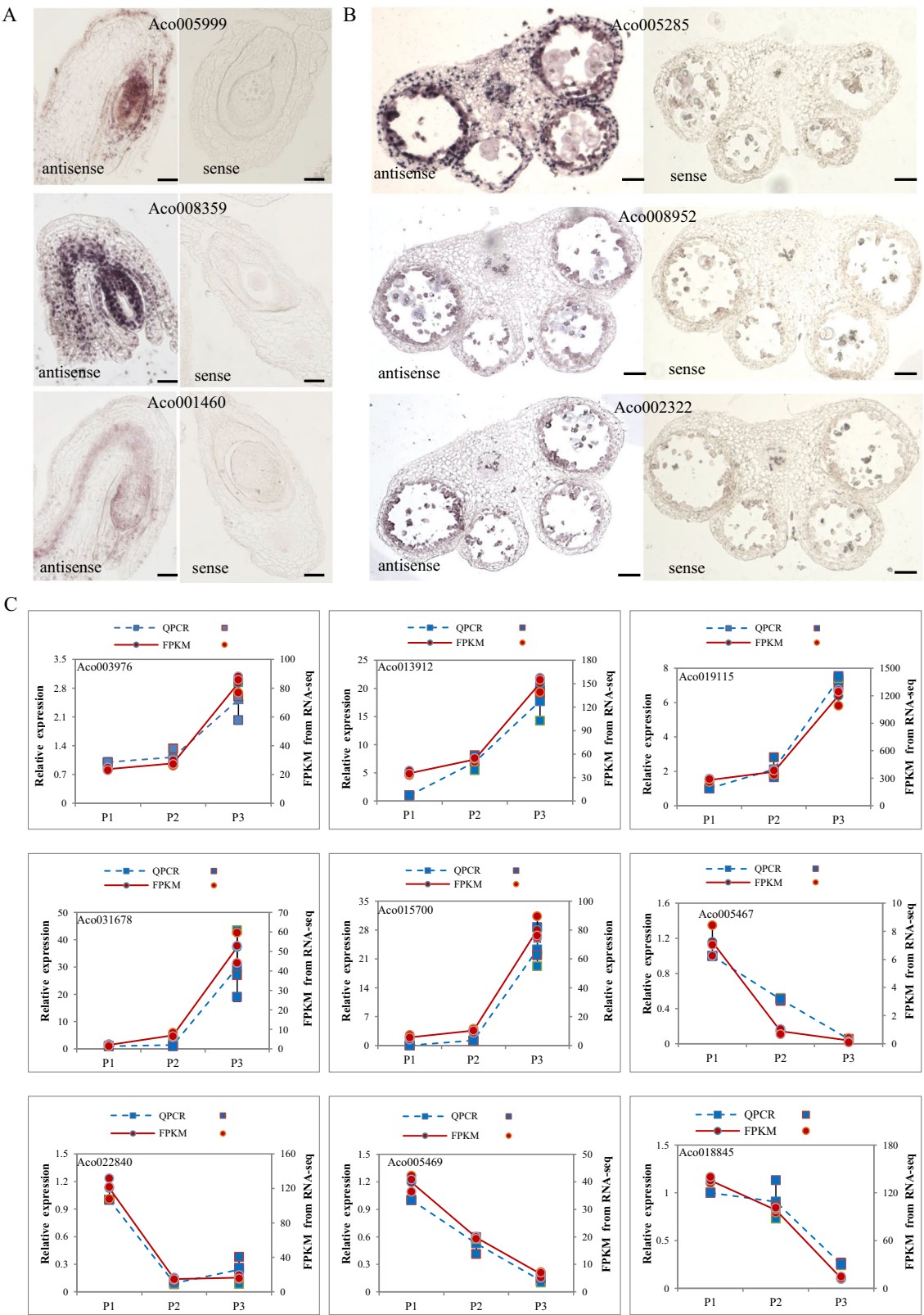

In this study, we generated 27 pineapple RNA-seq datasets for different floral samples and integrated them with nine published RNA-seq datasets, which included developing fruit stages. Both K-means clustering and WGCNA were used to analyze the RNA-seq data and identify tissue-specific gene clusters or modules, and many of these were identified by both methods. However, three small modules identified by WGCNA that were highly correlated

develops into a multiple fruit. Despite the economic significance of pineapple, relatively little study has been conducted on the molecular basis of pineapple floral organ formation and fruit development, which are crucial stages of the reproductive process. The sequenced pineapple genome and availability of transcriptomic data will make it possible to identify the key genetic factors that control these processes in pineapple.

**Fig. 6 In situ hybridization and RT-qPCR of Ovule 4-, Stamen 5-, and Petal-enriched genes. a** Longitudinal sections of pineapple stage 4 ovules with antisense and sense probes to detect Aco005999 (encoding chaperone protein ClpB), Aco008359 (encoding a MADS-box transcription factor), and Aco001460 (encoding a 70 kDa heat shock protein). Bars = 20 μm. **b** Cross-sections of pineapple stage 5 stamens with antisense and sense probes to detect Aco005285 (encoding a class I glutamine amidotransferase-like superfamily protein), Aco008952 (encoding an aquaporin-like superfamily protein), and Aco002322 (encoding a sequence-specific DNA binding transcription factor). Bars = 100 μm. **c** Validation of nine random selected petal specific subtilase genes by RT-qPCR. The left y-axis represents for the relative expression of RT-qPCR result and right y axis stands for the FPKM value from RNA-seq result. Blue dash line represents for RT-qPCR and red solid line represents for RNA-seq result. The letter P1–P3 means petal tissues at different stages.

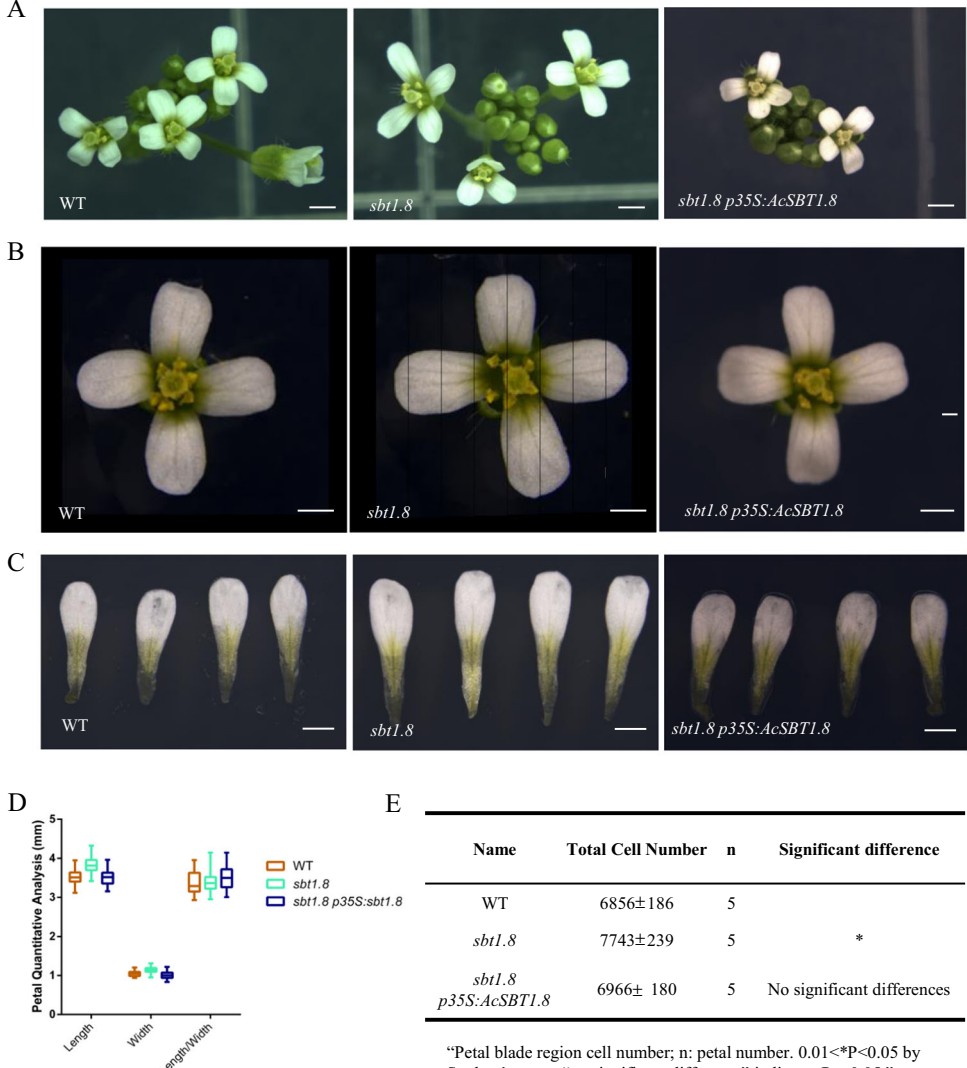

| Name | Total Cell Number | n | Significant difference |
|---|---|---|---|
| WT | 6856±186 | 5 | |
| sbt1.8 | 7743±239 | 5 | * |
| sbt1.8 p35S:AcSBT1.8 | 6966± 180 | 5 | No significant differences |

"Petal blade region cell number; n: petal number. 0.01<*P<0.05 by Student's t-test; "no significant difference" indicates P > 0.05."

**Fig. 7 Petal phenotype analysis of the *sbt1.8* mutant and the complementation lines. a–c** Inflorescence, flower and petal phenotypes of wild type (WT), *sbt1.8* and the *sbt1.8 p35S:AcSBT1.8* complementation lines at floral development stage 14. Bar = 1 mm. **d** Quantitative analysis of petal size in WT, *sbt1.8* and the complementation line. Values are given as the mean ± SD of 34 petals ($n = 34$) from independent plants (**$P < 0.05$ by Student's $t$ test). **e** Cell number in the petal blade region. Values are given as the mean ± SD of 5 ($n = 5$) petals from independent plants ($0.01 < *P < 0.05$ by Student's $t$ test; "no significant difference" indicates $P > 0.05$).

with Gynoecium 4, Ovule 4 and Petal 1 were not detected by the K-means method. WGCNA is a systems biology approach for describing the correlation patterns among genes across RNA-seq samples[24], and it can be used to find clusters (modules) of highly correlated genes. These methods have been successfully applied in various biological contexts, such as human cancer, mouse and yeast genetics[47–49]. Here, we used WGCNA to identify tissue- and stage-specific modules during pineapple reproductive development and the findings provide an important foundation for

further functional studies, which will significantly enhance our understanding of the genetic control of floral organ growth and fruit development in pineapple.

The ovule comprises the haploid female gametophyte and the surrounding diploid maternal sporophytic tissues. It is the site of double fertilization and subsequently develops into a seed[50]. Pineapple inflorescences consist of 50–200 individual flowers, each containing an inferior ovary with three carpels, and each carpel contains 20–60 ovules[5]. Pineapple flowers may therefore be

an ideal system to study ovule development. Here, we divided ovule development into seven stages. K-means clustering identified seven superclusters, the largest of which was supercluster 1, which included three clusters (C2, C3, and C5) and 13,480 genes. These genes had high expression levels at the different stages of ovule development, making supercluster 1 an ovule-specific supercluster. Of the 20 modules identified by WGCNA, the largest module (brown) was also associated with ovules. Interestingly, the lightcyan1 module was specifically associated with the Ovule 4 sample, and its hub genes included heat shock protein genes, indicating that this reproductive process is highly sensitive to stress. This hypothesis is consistent with previous reports that the reproductive (gametophytic) phase in flowering plants is highly sensitive to temperature stress[51]. Our previous transcriptomic analysis of female meiocytes also identified many stress response genes involved in ovule development[15].

The plum1 and yellow modules identified by WGCNA were highly correlated with Petal 1 and Petal 3, respectively. GO enrichment analysis revealed a large number of subtilisin-like serine protease genes in both modules. However, the subtilase genes in the Petal 1- and Petal 3-associated modules did not overlap, and they had opposite expression patterns over the course of petal development, indicating that these subtilase genes may undergo subfunctionalization. Subtilases belong to a superfamily of serine proteases and have a wide range of functions in different organisms. In fungi, subtilases act as cuticle-degrading proteases and play important roles in host invasion and nutrition acquisition[52,53]. In plants, subtilases have been associated with hypocotyl elongation, plant senescence, seed germination and stress resistance[54–57]. Using an *Arabidopsis* mutant, we demonstrated that *AcSBT1.8*, which was highly expressed at the Petal 3 stage, plays an important role in petal development likely by inhibiting the overproliferation of petal cells. Beyond this individual example of a petal development regulator, the present findings will help uncover the molecular mechanisms underlying flower organ and fruit development in pineapple.

## Methods

**Plant materials and tissue collection.** Pineapple (*Ananas comosus* L. Merr. var. MD-2) plants were grown in a greenhouse at 28 ± 1 °C under a 15-h-day/9-h-night cycle. We collected samples at different developmental stages. All tissues were dissected by hand and frozen immediately in liquid nitrogen, and a dissecting microscope was used for the ovule tissues. The tissues from at least three plants were combined to form one biological replicate, and three biological replicates were analyzed for each tissue.

For the analysis of T-DNA mutants, *Arabidopsis thaliana* Columbia-0 (Col-0) was used as wild type. The T-DNA insertion mutants *sbt1.8* (Salk-020799), *sbt5.3* (Salk-125788), and *sbt5.4* (Salk-025087) were obtained from the Arabidopsis Biological Resource Center. Polymerase chain reaction (PCR) amplification was used to confirm the presence of the T-DNA insertion in the *sbt1.8* mutant. *Arabidopsis* plants were grown at 22 °C under a 16-h-light/8-h-dark cycle.

**RNA extraction and library construction.** RNA was extracted using the RNeasy Plant Mini Kit (50) (Qiagen). A total of 1 μg RNA per sample was used for library preparation with NEBNext® Ultra™ RNA Library Prep Kit for Illumina. The cDNA ends were polished and ligated to NEB Next adapters. The adapter-ligated cDNA was amplified by 13 cycles of PCR, followed by purification using Agencourt AMPure XP Beads to obtain the final library for sequencing. The DNA yield and fragment insert size distribution of the library were determined on the Agilent Bioanalyzer system. The libraries were sequenced on a HiSeq2500 sequencing instrument using 150 bp paired-end protocols at the Center for Genomics and Biotechnology at Fujian Agriculture and Forestry University.

**RNA-seq analysis.** Raw reads were filtered by removing the adapter sequences and low quality sequences with Trimmomatic (v0.35)[58]. The parameters used in raw reads filter are set as: PE (for 27 of samples generated in this study)/SE (for 9 download samples), phred 33, LEADING:3, TRAILING:3, SLIDINGWINDOW:4:15, MINLEN:36. The published pineapple genome[12] was used as the reference genome. Clean reads were aligned to the reference genome with Tophat (v2.1.1)[59], and the alignment results were processed using Cufflinks (v2.2.1)[60] for gene quantification. We used two methods, Cuffdiff (v2.2.1) and

edgeR (v3.30.0), to analyze differential gene expression[60,61]. First, Cuffdiff was used to identify the differentially expressed genes (fold change ≥ 2, FDR ≤ 0.05) between any of the two samples among the 36 samples and then combine all the differentially expressed genes; at last removing the redundant ones. Second, we used featureCounts[62] to calculate read counts for each gene then used the edgeR program to detect differentially expressed genes (fold change ≥ 2, FDR ≤ 0.05) between any of the two samples among the 36 samples and then combine all the differentially expressed genes; at last removing the redundant ones. Finally, a total of 19,832 genes with FPKM values higher than 0.5 that were identified by both Cuffdiff and edgeR were used for further analysis.

**Gene network construction and visualization.** Co-expression networks were constructed using the WGCNA (v3.4.1) package in R[24]. The 19,832 differentially expressed genes identified by Cuffdiff and edgeR were used for the WGCNA unsigned co-expression network analysis; the average FPKM standardized by edger TMM was imported into WGCNA. The parameters used in the construction of WGCNA are set as: weighted network, unsinged; hierarchal clustering tree, Dynamic Hybrid Tree Cut algorithm; power 11; minModuleSize 30. The initial step of the co-expression analysis is to construct a matrix of pairwise correlations between all pairs of genes across all selected samples. Next, the matrix is raised to a soft-thresholding power based on the criterion of approximate scale-free topology and pickSoftThreshold function ($R^2 > 0.9$) to obtain an adjacency matrix. In the gene expression matrix, each row represents a gene and each column represents a specific tissue/stage, and the data in each grid represents the FPKM value of a specific gene in a specific tissue/stage sample. To identify modules of co-expressed genes, the topological overlap-based dissimilarity is constructed[63,64] then used as input for average linkage hierarchical clustering. Modules whose eigengenes are highly correlated are merged (mergeCutHeight = 0.25).

For the module-tissue association analysis, the eigengene value was calculated for each module and used to test the association with each tissue type. The total connectivity and intramodular connectivity (function softConnectivity), kME (for modular membership, also known as eigengene-based connectivity) and kME-*P* value were calculated for the 19,368 genes that were obtained after deleting the outliers (the non-tissues/stages-specific genes in the gray module), resulting in 20 tissue-specific modules. The networks were visualized using Cytoscape (v3.5.0)[65].

**K-means clustering.** To evaluate the robustness of the identified modules, we performed K-means clustering with Euclidean distance in MeV4.8[66], which yielded 21 clusters based on inputs of log2 relative FPKM values. Within these clusters, 15,872 genes with more obvious tissue- or stage-specific expression trends were selected to make the final 17 K-means clusters. TF genes and transcription coregulator (TC) genes were extracted from the respective clusters based on the Pineapple TFs and TCs annotation results[35]. An R pheatmap package and the log2 relative FPKM values of each TF and TC were used to generate heatmaps.

**GO enrichment analysis.** GO ontologies were assigned using bingo plug-in of Cycloscape software. The GO annotation file was download from the pineapple database (http://pineapple.angiosperms.org/pineapple/html/index.html). GO enrichment was derived with Fisher's exact test and a cutoff of false discovery rate less than 0.05; the genome annotation file described above was used as the reference. Only GO terms for Biological Process were shown. Redundant GO terms were removed with the ReviGO server (http://revigo.irb.hr/).

**Vector construction and plant transformation.** The predicted full-length *AcSBT1.8* (Aco003976) CDS sequence was obtained by searching the pineapple genomic database (http://pineapple.angiosperms.org/pineapple/html/index.html). The complete open reading frame (ORF) of *AcSBT1.8* was amplified from full-length cDNA clones by PCR with gene-specific primers (Supplementary Data 10). The PCR fragments were verified by DNA sequencing then cloned into the pENTR/D-TOPO vector (Invitrogen). The pENTR/D-TOPO clones were recombined into the destination vector pGWB605 using LR Clonase II (Invitrogen). The plasmid was extracted using the E.Z.N.A.® Plasmid Maxi Kit (Omega Bio-tek, Norcross, GA, USA) following the manufacturer's recommended procedure. An *Agrobacterium tumefaciens* strain (GV3101) carrying an *AcSBT1.8* CDS construct was introduced into *A. thaliana sbt1.8* mutant plants by the floral dip method. Transgenic plants were selected in soil by spraying with Basta (Basta: $H_2O$ = 1:1000). In total 13 independent T1 transgenic lines were used for phenotype observation.

**In situ hybridization and RT-qPCR.** For pineapple stage 4 ovules and stage 5 stamens, samples were fixed in RNase-free 4% (w/v) paraformaldehyde overnight (about 15 h) at 4 °C, dehydrated through an ethanol series, cleared through a xylene series, infiltrated through a paraffin series and finally embedded in 100% paraffin. Ten-micrometer sections were prepared and mounted on RNase-free glass slides. The fragments for probes were amplified with the primers listed in Supplementary Data 10. The DIGOXIN-labeled RNA antisense and sense probes were obtained as previously described[7,67]. Hybridization and immunological detection were performed according to the methodology described previously[68]. Images were captured using a ZEN 2011 microscope (Carl Zeiss, Germany).

Total RNA was extracted from petal tissues following manufacturer's protocol RNA extraction Kit (Omega Bio-Tek, Shanghai, China). cDNA was synthesized with the *EasyScript*® One-Step gDNA Removal and cDNA Synthesis SuperMix (Transgen, Beijing, China). RT-qPCR was conducted using TransStart® Top Green qPCR SuperMix (Transgen, Beijing, China). Actin2 was used as a reference gene (Supplementary Data 8). These assays were conducted for three biological replicates, and the results are shown as the mean ± standard deviations.

**Cell measurements and imaging of petal epidermal cells**. Petals from mature stage 14 flowers were dissected and stained in a solution containing 10 μg/mL propidium iodide for 1 h, followed by confocal imaging (Leica SP8) of the petal adaxial and abaxial epidermis. For each sample, more than six petals from three individual plants were analyzed. To observe the morphology of the petal epidermal cells, detached petals were directly examined under a TM-3000 table-top scanning electron microscope (Hitachi) equipped with a cooling stage. Petal blade cell length, cell width (distal region of the petal blade), cell area and conical cell angle were measured manually using Image J.

**Statistics and reproducibility**. For different developmental stages of pineapple flower tissues, at least three plants were combined to form one biological replicate, and three biological replicates were analyzed for each tissue. Petals were from mature stage 14 flowers of *Arabidopsis* and more than six petals from three individual plants were analyzed. The respective data for petal phenotype analysis are means ± SD of replications and statistical analyses shown in figures were performed using a unpaired Student's *t* test ($p < 0.05$ was considered statistically significant). The statistical analyses were performed with GraphPad Prism 6.0 (GraphPad Software Inc.).

**Reporting summary**. Further information on research design is available in the Nature Research Reporting Summary linked to this article.

## Data availability
The authors confirm that the data supporting the findings of this study are available within Supplementary Materials. The RNA-seq raw data of 27 samples generated in this study have been deposited in the European Nucleotide Archive (ENA) under accession number PRJEB38680. For the nine previously published pineapple RNA-seq datasets of Root, Leaf, Flower and Fruit were downloaded from https://de.iplantcollaborative.org/de/?type=data&folder=/iplant/home/cmwai/coge_data/Pineapple_tissue_RNAseq.

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

## Acknowledgements

We thank the Fujian Agriculture and Forestry University Center for Genomics and Biotechnology Sequencing Core Facility for sequencing the RNA-seq libraries. This work was supported by NSFC (U1605212 and 31970333 to Y.Q.), a Guangxi Distinguished Experts Fellowship, Science and Technology Major Project of Guangxi—Research and application of ecological and high efficient cultivation techniques for dominant and characteristic fruits (AA17204097-6), and Project of Guangxi featured fruit innovation team on pineapple breeding and cultivation post under national modern agricultural industry technology system (nycytxgxcxtd-17-05).

## Author contributions

L.W. genreated the RNA-seq libraries and analyzed the data. X.J. characterized the mutant. L.L. performed the in situ hybridization assay. Y.L., X.D., L.W., Y.-Q.L., D.L., and P.Z. analyzed the data. L.Z. and X.W. collected the materials for RNA-seq. Y.Q. conceived the study. L.W., Y.L., X.D., and Y.Q. wrote the paper.

## Competing interests

Yuan Qin is an Editorial Board Member for *Communications Biology*, but was not involved in the editorial review of, nor the decision to publish this article. The remaining authors declare no competing interests.
