## [Peer Review File · Communications Biology]

Reviewers' comments:

Reviewer #1 (Remarks to the Author):

In the present work, the authors have constructed co-expression gene networks in pineapple floral growth and fruit development using 36 RNA-seq datasets (including 27 generated in this study and 9 from publication). They identified tissue-specific modules as well as hub genes in ovule, stamen, petal and fruit development, with additional experimental validation. Based on these results, the authors investigated the role of AcSBT1.8 in petal development and found an increased petal size in the *sbt1.8* mutant. Overall the present work provides a rather important resource for functional analysis of floral organ growth and fruit development in pineapple. There are, however, some concerns that need to be clarified

1. As a resource paper, the raw data should be deposited in some public database.
2. In Figure 5B, what values used for the heatmap? From my reading, data points look like raw RPKM values. Since the gene expression of the selected genes may have different scales, it would be good to show scaled/normalized expression over samples for each gene so that it's more clear to see which tissues/stages the specific gene is dominantly expressed in.
3. In a recent work (DOI: 10.1038/s41467-018-06772-3), it has been shown that the SEP3-MIR319a-TCP4 pathway controlling petal development. Is there any evidence to show that the gene SBT1.8 is somehow related to this pathway?
4. There have been many and important works concerning gene networks controlling floral and fruit development in other model plants (*Arabidopsis*, rice and so on). It's not clear what novel message is provided in this work (compared to previous work). This should be clearly stated.

Reviewer #2 (Remarks to the Author):

The goals of this study were to generate tissue- and stage-specific transcriptomic profiles to identify modules and hub genes that may be involved in ovule, stamen, petal, and fruit development in pineapple. They comprehensively performed RNA-seq for 27 floral tissues at different developmental stages and combined these with 9 previously available samples. From these samples, they identified genes with differential expression using pairwise comparisons between samples. These genes were then clustered using both WGCNA and K-means methods to identify sets of genes correlated with the developing floral organs. Expression patterns were verified using in situ hybridization and the role of an identified subtilase gene for petal development was functionally verified in an *Arabidopsis* mutant.

Thank you to the authors for presenting an interesting study. These results will be found useful for others interesting in flower development and the application of co-expression clustering methods. The authors find a number of co-expression clusters that correlate with the different floral stages sampled and identify a number of candidate genes. These results were strengthened by the inclusion of in situ hybridization and mutant complementation.

In general, I found the manuscript to be written clearly and the methods employed seem appropriate. I don't find any major issues with the approach given that the goal of the study was the discovery of genes that may be involved with flower development in pineapple. The authors state that many of the

genes identified will be candidates for functional validation in future studies. Some clarification about how methods were employed and some justification for methodology need to be included. For instance, more explanation about how the differential expression tests were performed or what options were used for WGCNA. These and other thoughts are explained in my comments below.

A better presentation of the data should be done. Many of the supplemental tables are difficult to read because column names and row values are truncated. It would be better to provide these in other formats. Additionally, GO term enrichments should be summarized and redundant terms should be removed so that it is easier to make biological conclusions from them.

I appreciated that both WGCNA and K-means clustering was performed on the gene expression and it adds strength to the conclusions. However, I think more thorough comparisons should be made to identify whether the same (or similar) clusters are identified by both methods. It would strengthen the results to show that the gene clusters and membership within those clusters are similar despite the methodology chosen. I provide suggestions below.

Comments:

Line 93: What was the stage of development for the Flower sample?

Line 112-113: Describe the five expression categories. Why were five categories used and not some other number?

Line 114-116: These sentences don't add much because it is clear from Supplemental Figure 2A that the number of genes in each of the five expression categories is not appreciably different across all the samples.

Line 130: Unclear how the 19,832 genes were compiled. A more thorough explanation of how these 19,832 genes were obtained should be included in the methods. Please describe which comparisons were made, how these comparisons were performed in CuffDiff and edgeR, and whether up- or down-regulated genes (or both) were included.

Line 133: Describe the layout of the gene expression matrix. Here, or in the methods.

Line 136: Describe and justify the options used for WGCNA analysis.

Line 163: How was the GO enrichment analysis performed? What software and test were used? More details are needed.

Supplemental Table 5: Some column names and row values are truncated in this table. This appears to be an issue in other supplemental tables as well. Consider providing .xlsx or .csv files instead so that they can be properly opened and viewed.

In the brown module, for example, there are a lot of GO terms with enrichment that makes it difficult to parse the results. Consider running these lists through the ReviGO server (<http://revigo.irb.hr/>) to remove redundant GO terms. This would summarize these into smaller lists and make it easier to look for enriched GO categories.

Line 229: It is unclear why lightcyan1 (which is correlated with ovules) is placed in this section instead of with in the section above that describes brown and yellowgreen modules (also correlated with ovules). Talking about plum1 and yellow together here makes sense given that they are correlated to petals.

Line 259: I thought it was a really great idea to compare the clustering by WGCNA with K-means. More could be done here to describe and show how similar or different these methods performed for cluster identification. One idea might be to do a module preservation test, similar to what is done in WGCNA to compare two networks. It would provide quantitative answers to the question about whether these clusters are preserved (contain the same, or similar, sets of genes) between methods and would be fairly quick to implement. All that would be needed is the gene expression matrix and the gene cluster assignments from WGCNA and K-means. See here for an example: <https://horvath.genetics.ucla.edu/html/CoexpressionNetwork/ModulePreservation/Tutorials/HumanChimp.pdf>.

Line 276: How were these "superclusters" defined? How are they different from the 17 clusters identified above?

Lines 284-288: Several Arabidopsis orthologs are listed here for involvement in ovule development. Are these same genes found in the WGCNA clusters correlated to ovules?

Lines 308-320: Some references to the functional characterization of pineapple MADS-box genes might be needed in this section. For example: <https://www.mdpi.com/1422-0067/13/1/1039>, <https://link.springer.com/article/10.1186/s12864-019-6421-7>.

Line 362: It was a good idea to look at cell expansion versus cell proliferation to explain the very subtle phenotype of sbt1.8.

Methods: Please be sure to explicitly list which kits were used for RNA extraction and library construction. For software, include references and version numbers of the packages used. It is not stated anywhere how the GO enrichment tests were performed or what the corrected p-value threshold was for determining significance.

Line 480: How was read filtering and trimming performed? What were the criteria for trimming or removing low quality reads?

Line 486: Some clarification about how comparisons were performed is needed. Perhaps provide an example of a specific comparison that was made.

Line 496: Why was an unsigned network chosen (versus a signed network)? Describe the input data. What did rows and columns correspond to? What method of correlation was used (Pearson, Spearman, Midweight bicorrelation)? What was the R^2 cutoff used to determine the soft-thresholding power? Other options for WGCNA should be described here.

Line 506: The function is "softConnectivity".

Line 508: What explains the discrepancy between the 19,369 genes here and the original 19,832 genes input into WGCNA?

Line 535: How many independent transgenic lines were used?

Figure 5: There are no bootstrap values on the presented phylogenetic tree. What are the expression values in (B)?

In addition to the detailed questions/concerns/pointers from the reviewers, I would like to highlight a few questions that the authors could consider addressing them.

This includes how the genes enriched in different modules and tissues correlate with Arabidopsis and other plants' known phenotypes?

Response: Thank you for these points. As suggested, we have included more information about the functions of the orthologs of the pineapple tissue-specific genes in other species, such as Arabidopsis, rice, tomato and strawberry, in the revised manuscript, lines 196-201, lines 217-221, lines 254-258, and lines 319-324.

The additional questions are what the overall predictability of gene functions is; why the authors chose petal development over the fruit development (genes enriched in ovule) even though they chose seven stages of ovule development in their transcriptome study?

Response: Thank you for this point. Data presented in this study indicated that the subtilase genes were overrepresented in the eigengenes of Petal 1 and Petal 3 modules. Interestingly, there was no overlap between the Petal 1-associated and the Petal 3-associated subtilase genes, and the two groups of subtilase genes had opposite expression patterns throughout petal development. The significant enrichment of subtilase genes in petal-specific modules and the distinct expression patterns of the subtilase genes prompted us to investigate subtilase function in petal development. The functional study of ovule-specific genes would be carried out in another paper. We have explained this in the revised manuscript, lines 382-387.

Reviewers' comments:

Reviewer #1 (Remarks to the Author):

In the present work, the authors have constructed co-expression gene networks in pineapple floral growth and fruit development using 36 RNA-seq datasets (including 27 generated in this study and 9 from publication). They identified tissue-specific modules as well as hub genes in ovule, stamen, petal and fruit development, with additional experimental validation. Based on these results, the authors investigated the role of AcSBT1.8 in petal development and found an increased petal size in the sbt1.8 mutant. Overall the present work provides a rather important resource for functional analysis of floral organ growth and fruit development in pineapple. There are, however, some concerns that need to be clarified

1. As a resource paper, the raw data should be deposited in some public database.

Response: Thank you for this point. The raw data of 27 samples generated in this study have been deposited in the European Nucleotide Archive (ENA) under accession number PRJEB38680. The

9 previously published pineapple RNA-seq datasets of Root, Leaf, Flower and Fruit were downloaded from https://de.iplantcollaborative.org/de/?type=data&folder=/iplant/home/cmwai/coge_data/Pineapple_tissue_RNAseq. The information was provided in the revised manuscript, lines 638-643.

2. In Figure 5B, what values used for the heatmap? From my reading, data points look like raw RPKM values. Since the gene expression of the selected genes may have different scales; it would be good to show scaled/normalized expression over samples for each gene so that it's clearer to see which tissues/stages the specific gene is dominantly expressed in.

Response: Sorry for not providing this information. The $\log_2(\text{FPKM}+1)$ values were used for the heatmap. We added this information to the revised manuscript in Figure 5B and line 949.

3. In a recent work (DOI: 10.1038/s41467-018-06772-3), it has been shown that the SEP3-MIR319a-TCP4 pathway controlling petal development. Is there any evidence to show that the gene SBT1.8 is somehow related to this pathway?

Response: Thank you for this helpful comment. To detect whether *SBT* regulates petal development through this signaling pathway, we compared the expression levels of *SEP3*, *MIR319a* and *TCP4* in WT and *sbt1.8*. Comparable expression levels of these genes were detected in WT and *sbt1.8*, suggesting that *SBT* regulated petal development is likely to be independent of SEP3-MIR319/TCP4 signaling pathway. These results are provided in the revised manuscript, Supplemental Figure 7I and lines 411-417.

4. There have been many and important works concerning gene networks controlling floral and fruit development in other model plants (Arabidopsis, rice and so on). It's not clear what novel message is provided in this work (compared to previous work). This should be clearly stated.

Response: Thank you for this point. Revealing the spatio-temporal gene expressional profile along floral organ growth and fruit development helps understand the mechanism of reproductive development. Over the years, numerous studies have been implemented to detect the transcriptome profiling of developing petals, ovules, stamens and fruits/seeds in Arabidopsis thaliana, rice (*Oryza sativa*), soybean (*Glycine max*), tomato (*Solanum lycopersicum*), and maize (*Zea mays*). Although informative, these studies did not provide a complete set of spatio-temporal resolution of transcriptome data for the continuously developing reproductive organs. In this study, we performed transcriptome analysis for 27 different pineapple floral samples, which include three development stages of petal samples, four development stages of sepal samples, seven development stages of gynoecium samples, seven development stages of ovule samples and six development stages of stamen samples. In combination with previously published datasets for nine pineapple samples, including seven development stages of fruit samples, we performed weighted gene co-expression network analysis (WGCNA) and K-means clustering to identify network modules and tissue-specific gene clusters. The transcriptome and co-expression network analysis described in this study reports a comprehensive high spatio-temporal-resolution of genome-scale gene expression profiling, providing a foundation for the functional analysis of genes involved in

pineapple flower and fruit development. We have included this information in the revised manuscript, line 71-85, 93-96.

Reviewer #2 (Remarks to the Author):

The goals of this study were to generate tissue- and stage-specific transcriptomic profiles to identify modules and hub genes that may be involved in ovule, stamen, petal, and fruit development in pineapple. They comprehensively performed RNA-seq for 27 floral tissues at different developmental stages and combined these with 9 previously available samples. From these samples, they identified genes with differential expression using pairwise comparisons between samples. These genes were then clustered using both WGCNA and K-means methods to identify sets of genes correlated with the developing floral organs. Expression patterns were verified using in situ hybridization and the role of an identified subtilase gene for petal development was functionally verified in an Arabidopsis mutant.

Thank you to the authors for presenting an interesting study. These results will be found useful for others interesting in flower development and the application of co-expression clustering methods. The authors find a number of co-expression clusters that correlate with the different floral stages sampled and identify a number of candidate genes. These results were strengthened by the inclusion of in situ hybridization and mutant complementation.

In general, I found the manuscript to be written clearly and the methods employed seem appropriate. I don't find any major issues with the approach given that the goal of the study was the discovery of genes that may be involved with flower development in pineapple. The authors' state that many of the genes identified will be candidates for functional validation in future studies. Some clarification about how methods were employed and some justification for methodology need to be included. For instance, more explanation about how the differential expression tests were performed or what options were used for WGCNA. These are other thoughts are explained in my comments below.

A better presentation of the data should be done. Many of the supplemental tables are difficult to read because column names and row values are truncated. It would be better to provide these in other formats. Additionally, GO term enrichments should be summarized and redundant terms should be removed so that it is easier to make biological conclusions from them.

I appreciated that both WGCNA and K-means clustering was performed on the gene expression and it adds strength to the conclusions. However, I think more thorough comparisons should be made to identify whether the same (or similar) clusters are identified by both methods. It would strengthen the results to show that the gene clusters and membership within those clusters are similar despite the methodology chosen. I provide suggestions below.

Comments:

Line 93: What was the stage of development for the Flower sample?

Response: Thank you for this point. The Flower sample was published in the Nature Genetics (2015) paper and the information about the developmental stages of the flowers was not reported in that paper. To answer the reviewer's question, we contacted the authors of that paper and found out that the Flower sample was mixed flowers at different stages from multiple plants propagated clonally. The Leaf sample was the middle section of the youngest physiologically mature leaf, fourth from the apex. The Root sample was collected from mature plants. We have added this information in the revised manuscript, lines 106-108.

Line 112-113: Describe the five expression categories. Why were five categories used and not some other number?

Response: In this study, we divided all the expressed genes ($FPKM \geq 0.5$) into five categories according to a paper entitled as "Shifting the limits in wheat research and breeding using a fully annotated reference genome" published in 2018 "*Science*". We cited this literature in the revised manuscript, line 128.

Line 114-116: These sentences don't add much because it is clear from Supplemental Figure 2A that the number of genes in each of the five expression categories is not appreciably different across all the samples.

Response: Thank you for this point. We deleted these sentences.

Line 130: Unclear how the 19,832 genes were compiled. A more thorough explanation of how these 19,832 genes were obtained should be included in the methods. Please describe which comparisons were made, how these comparisons were performed in CuffDiff and edgeR, and whether up- or down-regulated genes (or both) were included.

Response: We are sorry for the unclear description. A total of 19,832 genes ($FPKM \geq 0.5$) that were identified as differentially expressed genes (fold change ≥ 2 , $FDR \leq 0.05$) between any of the two samples among the 36 samples by both of the Cuffdiff and edgeR methods were obtained and used for WGCNA analysis. We have explained this in the revised manuscript, lines 143-145 and lines 543-545.

Line 133: Describe the layout of the gene expression matrix. Here, or in the methods.

Response: In the gene expression matrix, each row represents a gene and each column represents a specific tissue/stage, and the data in each grid represents the FPKM value of a specific gene in a specific tissue/stage sample. The information was provided in the method section in the revised manuscript, lines 559-561.

Line 136: Describe and justify the options used for WGCNA analysis.

Response: Thank you for this point. The parameters used in the construction of WGCNA network are set as: weighted network, unsigned; hierarchical clustering tree, Dynamic Hybrid Tree Cut algorithm; power 11; minModuleSize 30. We clarified this in the revised manuscript, lines 552-558.

Line 163: How was the GO enrichment analysis performed? What software and test were used?
More details are needed.

Response: GO ontologies were assigned using bingo plug-in of CycloScope software. The GO annotation file was downloaded from the pineapple database (<http://pineapple.angiosperms.org/pineapple/html/index.html>). GO enrichment was derived with Fisher's exact test and a cutoff of false discovery rate less than 0.05; the genome annotation file described above was used as the reference. Only GO terms for Biological Process were shown. We have added the "GO enrichment analysis" in the methods section in the revised version, lines 583-590.

Supplemental Table 5: Some column names and row values are truncated in this table. This appears to be an issue in other supplemental tables as well. Consider providing .xlsx or .csv files instead so that they can be properly opened and viewed.

In the brown module, for example, there are a lot of GO terms with enrichment that makes it difficult to parse the results. Consider running these lists through the ReviGO server (<http://revigo.irb.hr/>) to remove redundant GO terms. This would summarize these into smaller lists and make it easier to look for enriched GO categories.

Response: Thank you for these points. As suggested, we provided the supplemental tables in excel files. We further analyzed GO term list through the ReviGO server (<http://revigo.irb.hr/>) to remove redundant GO terms in the revised version Supplemental Table 5.

Line 229: It is unclear why lightcyan1 (which is correlated with ovules) is placed in this section instead of with in the section above that describes brown and yellowgreen modules (also correlated with ovules). Talking about plum1 and yellow together here makes sense given that they are correlated to petals.

Response: Thank you for this helpful comment. We have replaced the lightcyan1 results to the ovule section and combined the plum1 and yellow modules in the petal section of the revised manuscript, lines 228-234 and lines 260-265.

Line 259: I thought it was a really great idea to compare the clustering by WGCNA with K-means. More could be done here to describe and show how similar or different these methods performed for cluster identification. One idea might be to do a module preservation test, similar to what is done in WGCNA to compare two networks. It would provide quantitative answers to the question about whether these clusters are preserved (contain the same, or similar, sets of genes) between methods and would be fairly quick to implement. All that would be needed is the gene expression

matrix and the gene cluster assignments from WGCNA and K-means. See here for an example: <https://horvath.genetics.ucla.edu/html/CoexpressionNetwork/ModulePreservation/Tutorials/HumanChimp.pdf>.

Response: Thank you for this great suggestion. We made a comparison between tissue-specific genes from WGCNA and K-means and revealed the consistency between the results from the two methods. For example, of the 4,157 eigengenes in ovule modules (brown and lightyellow) identified by WGCNA, 3,216 genes (77.4%) were also detected in the ovules-specific clusters (C2, C3 and C5) by K-means. Of the 1,559 eigengenes in stamen modules (grey60 and white) identified by WGCNA, 978 genes (62.6%) were also detected in the steman1-4 specific cluster (C1 and C11). We added these results in the revised manuscript, Supplemental Figure 6B and lines 295-302.

Line 276: How were these “superclusters” defined? How are they different from the 17 clusters identified above?

Response: Clusters with similar expression trends in the 17 clusters were further combined into seven superclusters. We added the definition of “superclusters” to the revised manuscript, lines 290-292.

Lines 284-288: Several *Arabidopsis* orthologs are listed here for involvement in ovule development. Are these same genes found in the WGCNA clusters correlated to ovules?

Response: Yes, these genes in the ovule specific supercluster 1 by K-means clustering are also identified as eigengenes in the ovule (brown) module by WGCNA. We clarified it in the revised manuscript. lines 322-323.

Lines 308-320: Some references to the functional characterization of pineapple MADS-box genes might be needed in this section. For example: <https://www.mdpi.com/1422-0067/13/1/1039>, <https://link.springer.com/article/10.1186/s12864-019-6421-7>.

Response: Thank you for this point. We had added references to the functional characterization of ABCE genes in *Arabidopsis* of the revised manuscript lines 350-352.

Line 362: It was a good idea to look at cell expansion versus cell proliferation to explain the very subtle phenotype of sbt1.8.

Response: Thank you for this kind comment.

Methods: Please be sure to explicitly list which kits were used for RNA extraction and library construction. For software, include references and version numbers of the packages used.

Response: Thank you for this point. We added the kits name and version numbers in the methods of the revised manuscript , lines 519-521.

It is not stated anywhere how the GO enrichment tests were performed or what the corrected p-value threshold was for determining significance.

Response: Thank you for this point. The GO enrichment analysis was assigned using bingo plug-in of Cycloscope software with Fisher's exact test and a cutoff of false discovery rate less than 0.05. We clarified it in the revised manuscript lines 583-590.

Line 480: How was read filtering and trimming performed? What were the criteria for trimming or removing low quality reads?

Response: Raw reads were filtered by removing the adapter sequences and low quality sequences with Trimmomatic (v0.3). The parameters used in raw reads filter are set as: PE (for 27 of samples generated in this study) /SE (for 9 download samples), phred 33, LEADING:3, TRAILING:3, SLIDINGWINDOW:4:15, MINLEN:36. We provided this information in the method section in the revised manuscript, line 531-533.

Line 486: Some clarification about how comparisons were performed is needed. Perhaps provide an example of a specific comparison that was made.

Response: Sorry for the unclear description. The comparisons were conducted between any of the two samples among the 36 samples. We revised the description in the methods of the revised manuscript, line 539 and lines 543-544.

Line 496: Why was an unsigned network chosen (versus a signed network)?

Describe the input data. What did rows and columns correspond to?

What method of correlation was used (Pearson, Spearman, Midweight bicorrelation)?

What was the R^2 cutoff used to determine the soft-thresholding power?

Other options for WGCNA should be described here.

Response: Thank you for these points. The "sign" represents the sign of weight on the edges. It represents positive or negative regulation between two nodes. In this study, the weight just represents the strength of relatedness between two nodes. We did not consider the positive or negative regulation between two nodes, so we choose "unsigned".

The input data of WGCNA is the gene expression profiles. In the gene expression matrix, each row represents a gene and each column represents a specific tissue/stage, and the data in each grid represents the FPKM value of a specific gene in a specific tissue/stage sample. We provided this information in revised manuscript lines 559-561.

Pearson correlation coefficient was used in the correlation analysis.

The R^2 cutoff used to determine the soft-thresholding power was set as $R^2 > 0.9$. We added the definition in revised manuscript lines 557-558.

The parameters used in WGCNA are set as: weighted network, unsigned; hierarchal clustering tree, Dynamic Hybrid Tree Cut algorithm; power 11; minModuleSize 30. This information has been added to the methods in the revised manuscript lines 552-554.

Line 506: The function is “softConnectivity”.

Response: We changed the “soft Connectivity” to “softConnectivity” in the revised manuscript line 567.

Line 508: What explains the discrepancy between the 19,369 genes here and the original 19,832 genes input into WGCNA?

Response: Thank you for this point. A total of 19,832 genes was used for the WGCNA unsigned co-expression network analysis. 464 genes were identified in the grey module with non-tissues/stages-specific expression pattern. After deleting these 464 outliers, 19,368 (19,369 has been corrected in lines 569) genes were obtained and used for the module-tissue association analysis. We clarified this information in the revised manuscript lines 569-571.

Line 535: How many independent transgenic lines were used?

Response: In total 13 independent lines were used for the phenotype observation. This information had been added to the revised manuscript, line 605.

Figure 5: There are no bootstrap values on the presented phylogenetic tree.

What are the expression values in (B)?

Response: Thank you for this point. The bootstrap values of the presented phylogenetic tree have been added to the tree in the revised version Figure 5A. The $\log_2(\text{FPKM}+1)$ values were used for the heatmap generation; we have added this information to Figure 5B in the revised version.

REVIEWERS' COMMENTS:

Reviewer #1 (Remarks to the Author):

Thank you to the authors for addressing my comments. I have no further comment.

Reviewer #2 (Remarks to the Author):

Thank you to the authors for taking into consideration the comments and suggestions from the reviewers and editor. After reading the revised manuscript, the authors have made important additions to explain the significance of the work and how they performed the experiments. In addition, the raw sequencing data have now been submitted to a public repository and the supplemental data is now in a more useful format.

In my opinion, there are no other major revisions that should be made. The authors have made the requested revisions and have produced a nice manuscript. The only additional suggestions I have for the authors on this manuscript are to check spelling and grammar.